# Gendered Parenting: Maternal Son Preference and Depressive Symptoms in Chinese Early Adolescents

**DOI:** 10.3390/bs14020104

**Published:** 2024-01-31

**Authors:** Fengqing Zhao, Yinge Wang, Yudan Li, Huifang Zhang, Sen Li, Zhongjie Wang, Jie Hou

**Affiliations:** 1School of Education, Zhengzhou University, Zhengzhou 450001, China; zhaofq@zzu.edu.cn (F.Z.); wangyinge096@gs.zzu.edu.cn (Y.W.); alice.kreep@gmail.com (H.Z.); 2School of Education, Hebei University, Baoding 071002, China; lisen19910227@foxmail.com; 3School of Politics and Public Administration, Zhengzhou University, Zhengzhou 450001, China

**Keywords:** maternal son preference, gendered parenting, parent–child attachment, school connectedness, depressive symptoms

## Abstract

This study focused on the nuanced phenomenon of gendered parenting by exploring how maternal son preference is associated with depressive symptoms among Chinese early adolescents. Focusing on 1093 junior high school students from a relatively affluent city in Western China, this study examined the mediating roles of mother–child attachment and father–child attachment as well as the moderating role of school connectedness in the relationship between maternal son preference and depressive symptoms. The results revealed a noteworthy positive correlation between maternal son preference and depressive symptoms among female adolescents, with no significant association observed in males. In addition, mother–child attachment and father–child attachment mediated the relationship between maternal son preference and girls’ depressive symptoms, supporting the “spillover effect” and “crossover effect”. Moreover, a moderation effect analysis indicated that a higher level of school connectedness can buffer the effect of maternal son preference on girls’ depressive symptoms, while a lower level of school connectedness can enhance the effect of maternal son preference on girls’ depressive symptoms. In addition, maternal son preference was negatively correlated with boys’ depressive symptoms in relation to high school connectedness. These insights help enhance people’s understanding of gendered parenting, emphasizing the enduring necessity of addressing son preference within the broader context of promoting gender equality.

## 1. Introduction

Son preference is a prevalent phenomenon across various cultures, with demographers observing a general preference for sons, whether in developed or developing countries. This inclination is especially pronounced in Asian countries such as South Korea and China [1]. Son preference reflects deeply ingrained cultural beliefs, notably within the context of Chinese clan culture. At the heart of Chinese clan culture lies the belief that male dominance is crucial to familial sacrifices and that men are the carriers of the family bloodline. This conviction reinforces the entrenched preference for sons over daughters, which leads to an imbalance in the boy/girl sex ratio at birth [2]. While the boy/girl sex ratio at birth in China had shown a downward trend in recent decades, it is still maintained at a high level, underscoring the enduring prevalence of son preference [3,4].

Son preference can result in higher mortality rates among women [5], prompt mothers to shorten the birth interval after having two daughters, and increase sex-selective abortion [6]. Despite these significant consequences, existing studies on son preference mostly focus on the phenomenon itself, its causes, and its broader influence on population and reproductive behavior. In this context, relatively scant attention has been given to its influence on individual mental health, especially that of adolescents, which highlights a critical gap in the current research landscape. Consequently, a thorough exploration of son preference within the Chinese cultural context and its association with adolescents’ mental health emerges as an imperative undertaking with far-reaching implications.

### 1.1. Maternal Son Preference and Adolescent Depressive Symptoms

Son preference reflects people’s understanding of the significance or value of having children of different genders, as well as motivational beliefs and value judgments when raising children of different genders [7]. It stems from the corrupt belief that a family must have male offspring or prefer male offspring when there are both male and female offspring and give sons more rights and care [8]. This preference reflects parents’ values regarding children and may influence parents’ attitudes toward and investment in children of different genders. Based on resource dilution theory, family resources are limited. When maternal son preference is high, mothers tend to sacrifice resources intended for their daughters and allocate more resources to their sons during resource allocation [9]. Previous studies supported this notion, indicating that sons tend to receive more family resources (e.g., food or care) during infancy and childhood to improve their chances of survival [10] and receive more educational opportunities and more attention from parents [11]. In addition, a few studies have shown concern for the negative consequences of son preference and indicate that son preference may decrease adolescent girls’ health [12] and even increase the likelihood of poor health, chronic disease, and depressive symptoms in their later lives [13,14]. This might be because son preference reduces family investment in girls’ health [12]. Particularly in families with limited resources, parental son preference often leads to adverse childhood experiences for daughters which, in turn, are linked to poor health in later life, depressive symptoms, and anxiety [14].

According to the ecological systems theory, family is the most important microenvironment for adolescents as it is the immediate setting of adolescents’ daily interactions and relationships, and it exerts a direct and profound impact on adolescents’ development [15]. Therefore, parents’ beliefs and attitudes toward their children directly affect their growth and development. Social comparison theory holds that individuals gain self-evaluation in comparison with others [16]. Adolescents tend to compare how their parents treat them to how their parents treat their siblings, and through these comparisons, they form a sense of identity and self-worth [17]. Comparisons between siblings tend to lead to feelings of injustice, anxiety, and insecurity, and children of lower status tend to have a lower sense of self-worth, which may lead to depression [18,19].

However, only some studies supported a negative correlation between parental preference and depressive symptoms in favored adolescents [14]. Some studies even indicate that parental preference is significantly positively related to the depressive symptoms of favored adolescents [20]. Therefore, the relationship between maternal son preference and male adolescent depressive symptoms needs to be further explored. In addition, previous studies did not consider the unique effect of maternal son preference on the development of adolescents across different genders [12,14]. As the main caregiver in the family, the mother spends more time with the children and has a more direct influence on adolescents’ well-being. In addition, the mother herself may be the victim of son preference in her native family, and the gender-based differential treatment that women may encounter at all stages of their life and the resulting sense of inequality between men and women will promote them to form the concept of son preference [21]. Therefore, it is necessary to explore the association between maternal son preference and adolescent depressive symptoms in different genders.

### 1.2. The Mediating Role of Parent–Child Attachment

Parental son preference may be a risk factor for adolescent girls’ parent–child attachment. According to attachment theory, a parent–child attachment is formed in the process of interaction between a child and their parents and is a deep and continuous emotional connection [22]. Positive parent–child interactions (e.g., parents caring for their children and encouraging children to be autonomous) are conducive to the formation of the parent–child relationship and subsequently forms a secure attachment [23]. Earlier longitudinal studies of twins consistently showed that when the pattern of interaction between a mother and her favored child was more sensitive, the twin would develop a secure attachment with the mother. However, unfavored children were more likely to develop insecure attachments, anxiety, and low self-esteem [24,25]. A recent cross-sectional study also showed that parents’ differential treatment may interfere with children’s ability to develop secure attachment relationships [26].

Additionally, it is important to distinguish the effects of maternal son preference on girls’ attachments with their fathers and mothers. Based on the family system theory, the family is composed of multiple subsystems, including the marital subsystem, parent–child subsystem, and sibling subsystem, and the overall function of the family system is the result of the interactions between multiple subsystems [27]. Therefore, maternal son preference will not only directly influence the mother–son relationship but also the mother–daughter relationship characterized by the “spillover effect” and the father–daughter relationship characterized by the “crossover effect” [28]. According to the spillover effect, the mother’s cognition, emotion, or behavior in the mother–son subsystem will affect these aspects in the mother–daughter subsystem [29]. According to the crossover effect, the cognition, emotion, or behavior of mother in the mother–son subsystem can affect the cognition, emotion, or behavior of father in the father–child subsystem [30]. That is to say, the mother’s cognition, emotion, and behavior in the mother–son subsystem will not only affect the mother–child subsystem but also the father–child subsystem.

Parent–child attachment is closely related to the development of depressive symptoms in adolescents. Studies have consistently indicated that parent–child attachment is still an important predictor of adolescent depressive symptoms, and good-quality parent–child attachment is a protective factor for reducing adolescent depressive symptoms [31]. Good parent–child attachment can largely alleviate internalizing problems, such as depression and anxiety, in children and adolescents, while poor parent–child attachment may cause children to perceive the world as indifferent and foster a cold, unfriendly attitude in interpersonal communication, leading to symptoms related to depression [32]. A multilevel meta-analysis suggested that insecure attachment to a primary caregiver was associated with the development of depressive symptoms and that insecure attachment had a greater impact on internalizing problems in girls than boys [31]. A longitudinal study also showed that poor parent–child attachment predicted the development of depressive symptoms among children and adolescents, even after accounting for all time-invariant factors [33].

Although previous studies confirmed that the quality of the parent–child attachment is significantly related to children’s depressive symptoms, there may be differences in the relationship between the father–child attachment and the mother–child attachment and adolescent depressive symptoms. On one hand, fathers can help children develop social capability and autonomy, and high levels of social capability and autonomy can prevent adolescents from developing depressive symptoms [34]. For example, a cross-sectional study found that the father–child attachment was more strongly associated with depressive symptoms in children and adolescents than the mother–child attachment [34]. On the other hand, children rely more on their mothers as a “safe haven” in times of distress than on their fathers [35]. A longitudinal study showed that the individual’s attachment to the mother was more important, and the mother–child attachment was more strongly associated with children’s social development [33]. A recent cross-sectional study also found that only the correlation between the mother–child attachment and depressive symptoms was significant [36]. Thus, there may be differences in the effects of the mother–child attachment and father–child attachment on depressive symptoms in adolescents.

Maternal son preference may lead to poor mother–daughter attachment which, in turn, is associated with depressive symptoms in female adolescents. When maternal son preference is high, a mother may be more likely to interact with sons and ignore daughters in the family, which will cause daughters to experience less maternal emotional warmth and more emotional neglect and develop insecure attachment relationships [25,37]. This insecure attachment relationship makes them less likely to seek support and comfort from others and learn less about other people’s solutions to difficulties when they encounter difficulties [38]. Thus, it is difficult to adapt to negative events and easy to fall into a sense of helplessness and powerlessness, which ultimately leads to depressive symptoms [39,40]. Therefore, the mother–child attachment and father–child attachment may mediate the association between maternal son preference and adolescent depressive symptoms.

### 1.3. The Moderating Role of School Connectedness

During the transitional phase of middle school, adolescents experience a gradual detachment from their family environments and spend less time interacting with their families and more time within the school environment [41]. In this context, schools emerge as pivotal settings that affect their cognitive and socio-emotional development [42]. School connectedness refers to the level of support students receive from their peers and teachers and the degree to which they feel they belong to the school [43]. It plays a crucial role as a protective factor for adolescents’ externalizing behaviors (e.g., addiction, aggression) and internalizing problems (e.g., depression, anxiety) [44]. Empirical studies have demonstrated a correlation between school connectedness and mental health [45], and less school connectedness in early adolescence predicts depressive symptoms in adolescents six months later [46].

Although studies have shown that maternal son preference and poor parent–child attachment affect the occurrence and development of depressive symptoms in adolescents, not all adolescents who experience these problems will develop depressive symptoms, which may be moderated by school connectedness. According to the ecological systems theory, the family and school environments are two crucial microsystems. The interaction between these two microsystems forms a mesosystem, and both the microsystems and mesosystem collectively influence the development of adolescents [15,47]. Adolescents who are more connected to school can receive more social support from teachers and peers and experience a heightened sense of emotional belonging [48]. Therefore, the adverse effects of inadequate parental care in the family on adolescents can be alleviated to some extent by a positive school environment. For instance, cross-sectional evidence indicates that a high level of school connectedness can buffer the association between parental neglect and problematic behavior such as dependence on mobile short videos among left-behind adolescents [49]. Additionally, high school connectedness can buffer the association between insecure attachment and adolescents’ depressive symptoms and suicidality, enhancing the protective role of secure attachment against adolescents’ depressive symptoms and suicidal tendencies [50]. Longitudinal evidence also supports the idea that school connectedness moderates the negative impact of parent–child conflict on adolescent depression one year later [43].

### 1.4. The Current Study

Based on the above argument, there still exist limitations and gaps in the existing literature. First, previous studies were mostly concerned with the general family son preference and seldom focused on the maternal son preference, not mentioning the association between maternal son preference and adolescents’ psychological well-being. Second, most of the previous studies, situated within the domains of demography and sociology, mostly adopted qualitative methodologies which may not reveal the full picture of the phenomenon of son preference. Thus, researchers are encouraged to use quantitative methods on the basis of a relatively large sample to draw a more precise conclusion. Third, a gap exists in that the existing studies did not consider the interactions within family subsystems, nor did they address the interaction of the family system and school system in adolescents’ development to reveal the underlying mechanism of the effect of son preference on adolescent depressive symptoms. Therefore, based on the family system theory, the internal work model of attachment, and the ecological system theory, this study examines the relationship between maternal son preference and depressive symptoms among adolescents, as well as the roles of parent–child attachment and school connectedness in their association, and three hypotheses are proposed as follows. Since previous studies on maternal son preference and depressive symptoms in male adolescents have not reached consistent conclusions, we do not make specific hypotheses.

**H1.** 
*Maternal son preference will significantly associate with adolescent depressive symptoms. Specifically, maternal son preference will be positively and significantly associated with depressive symptoms among female adolescents (H1a); in addition, we expect to explore the association between maternal son preference and adolescent depressive symptoms among male adolescents (H1b).*


**H2.** 
*Mother–child attachment and father–child attachment will mediate the relationship between maternal son preference and adolescent depressive symptoms (H2).*


**H3.** 
*School connectedness will moderate the mediating model between maternal son preference, parent–child attachment, and depressive symptoms (H3).*


## 2. Method

### 2.1. Participants

This study was conducted with a convenience sample of junior high school students. The school is a private school located in Xi’an City, a relatively rich city in the midwestern area of China. It covers an area of 500 acres. A total of 1111 questionnaires were collected from 21 first-grade classes. After excluding the questionnaires with high missing data rates (>15%), the number of final valid questionnaires was 1093, and the recovery rate was 98.38%. The adolescents ranged in age from 12 to 14 (M = 12.96, SD = 0.42), and 42.26% were girls. In addition, 89.66% of the participants lived in urban areas, 7.22% of the participants lived in rural areas, and data were missing for 3.12%. Regarding annual household income, 7.6% reported less than CNY 30,000, 8.05% reported between CNY 30,000 and 60,000, 18.3% reported between CNY 600,000 and 100,000, 61.3% reported above CNY 100,000, and data were missing for 4.76%. The overall subjective socioeconomic level of the family is above average (range = 1–10, M = 6.81, SD = 1.62) [51].

### 2.2. Measures

#### 2.2.1. Maternal Son Preference

This study used three items investigating preferences for boys from the Value of Child Scale (VOC) [7,13], such as “If your mother did not have any sons, how much did she regret this?” According to the scoring standard, the items were scored from 0 to 3, and higher scores indicated a mother’s greater preference for boys. In this study, the internal consistency coefficient for maternal son preference was 0.718.

#### 2.2.2. Parent–Child Attachment

This study used two subscales of father–child attachment and mother–child attachment in the Inventory of Parent and Peer Attachment (IPPA) [52], such as “I tell my mother/father about my problems and troubles”. Each subscale contains 10 items which are divided into three dimensions: communication, trust, and alienation [36,53]. The scale is scored from 1 to 5 (from completely inconsistent to completely consistent). The higher the score, the higher the level of attachment security. In this study, the internal consistency coefficients of father–child attachment and mother–child attachment were 0.795 and 0.771, respectively.

#### 2.2.3. School Connectedness

The School Connectedness scale includes 10 items which are divided into three dimensions, peer support, teacher support, and school belongingness [54,55], such as “I can rely on my classmates when facing difficulties”. Participants answer on a scale from 1 to 5 (from completely disagree to completely agree); items 2 and 5 are reverse-scored items. In addition, the higher the total score, the higher the individual’s level of school connectedness. In this study, the internal consistency coefficient of school connectedness was 0.871.

#### 2.2.4. Depressive Symptoms

This study used the Simplified Chinese Version of the Depression–Anxiety–Stress Scale (DASS-21) [56,57], such as “I found it hard to wind down”. The scale has 21 items and is divided into three dimensions: depressive symptoms, anxiety, and stress. We only measured the depressive symptom dimension. The scale is scored from 0 to 3 (from completely inconsistent to completely consistent), with higher scores indicating a more intense depressive emotional experience. According to the classification standard, the score of the depressive symptom subscale was multiplied by 2 to make a grade classification. The scores 10, 14, and 21 indicate critical values of mild, moderate, and severe depressive symptoms, respectively. The internal consistency coefficient of the depressive symptom dimension was 0.876.

### 2.3. Procedure

In this study, professionally trained postgraduate psychology students served as the main examiners. After receiving informed consent from the school, students, and parents, we invited the students to participate in the investigation voluntarily during class as a unit. Before the students filled out the questionnaire, the examiner provided a brief explanation of the instructions and emphasized the key points to note. The students filled out the questionnaires in their classroom and took them back on the spot after completing them. Finally, each participant received a small gift to motivate them to participate.

### 2.4. Statistical Analyses

Because maternal son preference may have different correlation with adolescent boys’ and girls’ psychological indicators, this study took gender as a benchmark and divided the participants into male and female groups for correlation analysis, mediation effect analysis, and moderation effect analysis. First, descriptive statistics and zero-level correlation analyses were performed for all study variables (established from the mean of items) using SPSS 26.0 software. Next, the PROCESS procedure (model 4) was used to analyze the mediating effect, with a bootstrap sampling number of 5000. When the confidence interval of 95% did not contain 0, it indicated that there was a significant mediating effect or moderating effect. Finally, the PROCESS procedure (model 59) was used to analyze the moderating effect of school connectedness.

A multicollinearity diagnosis was performed on the selected variables before the analysis [58]. The results showed that the tolerance value ranged from 0.49 to 0.99 (all greater than the critical value of 0.2), and the variance inflation factor (VIF) ranged from 1.01 to 2.04 (all less than the critical value of 5), so there was no multicollinearity problem. Since all the variable data were self-reported by the participants, a common method bias may result. A Harman single-factor test was used to test the common method bias of the date [59]. An exploratory analysis of the items in the questionnaire showed that the variation rate of the first factor was 29.81%, which was lower than the critical value (40%). Therefore, this study was not affected by a severe common method bias.

## 3. Results

### 3.1. Descriptive Statistics and Correlation Analysis of Main Variables

In this study, 48.21% of adolescents reported a maternal son preference score greater than 0, including 36.96% male adolescents and 11.25% female adolescents. This shows that the male adolescents reported more maternal son preference than the female adolescents. In addition, the detection rate for depressive symptoms was 16.38%, of which mild depressive symptoms, moderate depressive symptoms, and severe depressive symptoms accounted for 6.13%, 6.13%, and 4.12%.

The correlation analysis results are shown in Table 1 and Table 2. In the female group, maternal son preference was positively correlated with depressive symptoms (*p* < 0.01) and negatively correlated with mother–child attachment, father–child attachment, and school connectedness (*p <* 0.01). In the male group, maternal son preference was not significantly correlated with depressive symptoms, mother–child attachment, or father–child attachment (*p* > 0.05), and it was only negatively correlated with school connectedness (*p* < 0.05). In addition, in both groups, depressive symptoms were negatively correlated with mother–child attachment, father–child attachment, and school connectedness (*p <* 0.01); mother–child attachment was positively correlated with father–child attachment, and both mother–child attachment and father–child attachment were positively correlated with school connectedness (*p* < 0.01).

### 3.2. The Mediating Effect of Parent–Child Attachment between Maternal Son Preference and Depressive Symptoms

We further explored the mediating role of father–child attachment and mother–child attachment in the relationship between maternal son preference and adolescent depressive symptoms. Among male adolescents, the mediating effects of father–child attachment (95% CI = [−0.120, 0.024]) and mother–child attachment (95% CI = [−0.064, 0.050]) on the relationship between maternal son preference and adolescent depressive symptoms were not significant.

Among female adolescents, after adding mediating variables, maternal son preference was still positively correlated with depressive symptoms (*β* = 0.090, *t* = 2.291, *p* < 0.05). This showed that the direct effect of maternal son preference on depressive symptoms was significant after controlling for parent–child attachment. In addition, maternal son preference was negatively correlated with mother–child attachment (*β* = −0.156, *t* = −3.392, *p* < 0.01) and father–child attachment (*β* = −0.237, *t* = −5.226, *p* < 0.001); both mother–child attachment (*β* = −0.371, *t* = −6.862, *p* < 0.001) and father–child attachment (*β* = −0.222, *t* = −4.027, *p* < 0.001) were negatively correlated with depressive symptoms. This showed that the indirect effects of mother–child attachment and father–child attachment were significant. In conclusion, the results showed that mother–child attachment and father–child attachment played partial mediating roles between maternal son preference and depressive symptoms (Figure 1). Among them, the direct effect accounted for 44.8%, and the indirect effect accounted for 28.9% and 26.3%, respectively (Table 3).

### 3.3. The Moderating Effect of School Connectedness

Since the association between maternal son preference and depressive symptoms was significant in the female group (*p* < 0.01) and parent–child attachment played a mediating role between maternal son preference and depressive symptoms in female adolescents, we further tested the effect of the interaction of maternal son preference, parent–child attachment, and school connectedness on adolescents’ depressive symptoms among the female group. The results showed (Table 4) that only the interaction of maternal son preference and school connectedness was significantly negatively correlated with depressive symptoms (*β* = −0.164, *t* = −4.829, *p* < 0.001). A simple slope analysis using the pick-a-point approach showed that maternal son preference was positively correlated with depressive symptoms at a low level of school connectedness (*β* = 0.194, *t* = 4.456, *p* < 0.001) and negatively correlated with depressive symptoms at a high level of school connectedness (*β* = −0.134, *t* = −2.394, *p* < 0.05). The results suggested that school connectedness played a moderating role in the relationship between maternal son preference and girls’ depressive symptoms.

In order to understand the boundary value of the moderating effect of school connectedness and the value range of school connectedness when the simple slope was significantly less than 0, we used the Johnson–Neyman technique to test the simple slope [60]. The results (Figure 2) showed that when school connectedness was less than or equal to −0.25 standard deviations, maternal son preference was positively correlated with depressive symptoms. In addition, the association of maternal son preference and depressive symptoms was weakened with an increase in school connectedness. When school connectedness was greater than or equal to 0.79 standard deviations, maternal son preference was negatively correlated with depressive symptoms, and the influence of maternal son preference on depressive symptoms increased with an increase in school connectedness.

Among male adolescents, the mediating effect of parent–child attachment on maternal son preference and depressive symptoms was not significant. Therefore, we only investigated the moderating effect of school connectedness on maternal son preference and depressive symptoms. The results showed that maternal son preference was not significantly associated with depressive symptoms (*β* = −0.013, *t* = −0.362, *p* > 0.05), school connectedness was negatively correlated with depressive symptoms (*β* = −0.461, *t* = −12.949, *p* < 0.001), and the interaction of maternal son preference and school connectedness could significantly negatively correlate with depressive symptoms (*β* = −0.087, *t* = −2.404, *p* < 0.05). A simple slope analysis using the pick-a-point approach showed that maternal son preference was not significantly correlated with depressive symptoms at a low level of school connectedness (*β* = 0.074, *t* = 1.418, *p* > 0.05) but was negatively correlated with depressive symptoms at a high level of school connectedness (*β* = −0.094, *t* = −1.999, *p* < 0.05). The results suggested that school connectedness played a moderating role in the relationship between maternal son preference and boys’ depressive symptoms.

We further used the Johnson–Neyman technique to explore the boundary value, and the results (Figure 3) showed that when school connectedness was less than or equal to −2.06 standard deviations, maternal son preference was positively correlated with boys’ depressive symptoms. However, when school connectedness was greater than or equal to 0.89 standard deviations, maternal son preference was negatively correlated with boys’ depressive symptoms, and the association of maternal son preference with depressive symptoms increased with an increase in school connectedness.

## 4. Discussion

This study aimed to explore the relationship between maternal son preference and depressive symptoms among Chinese early adolescents while considering the roles of father–child attachment, mother–child attachment, and school connectedness. The results showed that maternal son preference was positively correlated with depressive symptoms only among female adolescents; father–child attachment and mother–child attachment played mediating roles between maternal son preference and depressive symptoms among female adolescents; and school connectedness moderated the relationship between maternal son preference and depressive symptoms among female adolescents.

### 4.1. The Relationship between Maternal Son Preference and Adolescent Depressive Symptoms

This study found that maternal son preference was positively correlated with depressive symptoms only among female adolescents, which supports Hypothesis 1. This may be explained by two aspects. First, maternal son preference reduces equal resource allocation among siblings, and adolescents may have depressive symptoms when they perceive an unjust allocation of resources. Resource dilution theory holds that family resources are limited. When the level of maternal son preference is high, the resources in the family are more inclined toward boys, which reduce the family resources received by girls [61], such as time resources and educational resources [62]. For girls, the perceived deprivation of deserved resources can evoke negative emotions such as sadness, anger, and shame, leading to depressive symptoms over time. In contrast, boys receive the resources they deserve, so the level of maternal son preference is not related to their depressive symptoms [63]. Second, maternal son preference leads to girls’ low self-evaluation, which is a core element of depressive symptoms. Social comparison theory holds that individuals self-evaluate in comparison with others [16]. Adolescents frequently engage in comparisons of their mothers’ treatment of siblings and themselves. In families with higher levels of maternal son preference, girls may think that they receive less warmth and support from their mothers, which leads to negative self-evaluation, low self-esteem, and high depressive symptoms [64,65].

### 4.2. The Mediating Role of Parent–Child Attachment

This study found that among female adolescents, mother–child attachment and father–child attachment played mediating roles in the relationship between maternal son preference and depressive symptoms, which partially supports Hypothesis 2. In situations in which maternal son preference is pronounced, adolescent girls tend to report low levels of attachment with both their mothers and fathers; these low levels of attachment are associated with more severe depressive symptoms. According to attachment theory, the parent–child attachment forms an individual’s internal working model of the self and others. If the attachment between girls and their parents is poor, it can foster a perception that their parents are unsupportive and reject them, leading to a negative internal working pattern regarding the self (I am worthless) and others (no one cares about me; others are not available to me), which may increase the risk of depressive symptoms [66,67].

First, the mother–child attachment mediated the association between maternal son preference and girls’ depressive symptoms, which supported the “spillover effect model”. On one hand, when a mother has a higher preference for sons, her desire to bear sons is usually higher. When the actual birth results in a daughter, the difference between this expectation and reality may evoke negative emotions or behaviors, consequently diminishing communication with daughters and leading to a weakened mother–child attachment [13]. This causes daughters to rarely seek help from their mothers when facing problems or difficulties, which increases their risk of depressive symptoms [68]. On the other hand, mothers’ differential parenting styles may affect children’s perception of maternal emotional warmth, and adolescents who perceive less emotional warmth may develop depressive symptoms. When the maternal son preference is higher, girls may experience disappointment and become angry with their mothers due to their mothers’ blind preference for sons. This can lead to a diminished sense of emotional warmth and an alienated mother–child relationship, fostering insecure mother–child attachment [26]. Insecure mother–child attachment, in turn, heightens the likelihood of girls experiencing more negative emotions, such as depressive symptoms [69].

Second, the father–child attachment mediated the association between maternal son preference and adolescent girls’ depressive symptoms, which supported the “crossover effect model”. This could be attributed to the fact that maternal son preference is not just the mother’s personal preference but also reflects the beliefs and values held by the whole family culture. In Chinese traditional culture, mothers often hold the belief that having a son is crucial for gaining acceptance within the family and proving their worth, so maternal son preference reflects the values of gender inequality within the family [70]. Therefore, when mothers prefer sons, family resources (e.g., health and education) are more inclined to be allocated to sons, which reduces the parents’ provision of love to girls, thus affecting the attachment relationships between girls and her parents [10]. In such a family, girls tend to experience negative emotions such as loneliness, sadness, grievance, anger, and shame and gradually form the belief that they are not good enough and are unworthy of being loved, which increases the risk of depressive symptoms.

### 4.3. The Moderating Effect of School Connectedness

In female adolescents, there was a positive correlation between maternal son preference and depressive symptoms in the case of a low level of school connectedness and a negative correlation between maternal son preference and depressive symptoms in the case of a high level of school connectedness. Furthermore, using the Johnson–Neyman technique, it was found that when school connectedness was less than or equal to −0.25 standard deviations, maternal son preference was positively correlated with depressive symptoms. When school connectedness was greater than or equal to 0.79 standard deviations, maternal son preference was negatively correlated with depressive symptoms. In male adolescents, maternal son preference was not correlated with depressive symptoms at a low level of school connectedness but was negatively correlated with depressive symptoms at a high level of school connectedness. Furthermore, using the Johnson–Neyman technique, it was found that when school connectedness was greater than or equal to 0.89 standard deviations, maternal son preference was negatively correlated with depressive symptoms.

When the school connectedness was below −0.25 standard deviations, the depressive symptoms of female adolescents increased significantly with an increase in maternal son preference. This can be explained from the “reinforcement model” in that one relationship can enhance the effect of another relationship on problem behavior. When both relationships are good, they can mutually reinforce each other, amplifying their positive influence on reducing problem behavior; when both relationships are poor, they can enhance each other’s negative influence on increasing problem behavior [50,71]. Therefore, the combination of high maternal son preference and low school connectedness makes female adolescents experience more depressive symptoms.

When the school connectedness was higher than 0.79 standard deviations, the depressive symptoms of female adolescents decreased significantly with an increase in maternal son preference. This can be explained using the “buffering model”; when one relationship is poor, another good relationship can buffer the negative effects of the poor relationship [71]. In a positive school environment, female adolescents can seek support from peers and teachers to fulfill emotional needs that may be unfulfilled by family. This support can act as a compensatory mechanism, alleviating the negative impact of an unfavorable family environment on their development and ultimately reducing the likelihood of depressive symptoms [72]. Additionally, when school connectedness was higher than 0.89 standard deviations, the depressive symptoms of male adolescents decreased significantly with an increase in maternal son preference. This indicates that school connectedness is one of the important factors in the positive development of adolescents [73]. A high level of school connectedness can satisfy the emotional and relational needs of adolescents and stimulate their internal motivation to seek positive development [74].

As an important protective factor, school connectedness can buffer the negative impact of maternal son preference on female adolescents. Therefore, with respect to reducing the negative impact of a lack of maternal care on female adolescents, we can compensate by providing more school connectedness for them. First, in enhancing peer support, teachers and schools can organize peer support groups to promote interaction and cooperation among students so that adolescents can seek support from their classmates when they encounter difficulties [75]. Second, in enhancing teacher support, teachers should adopt supportive communication methods and pay timely attention to students’ individual needs so as to establish positive teacher–student relationships and enhance students’ trust in teachers [76]. Third, schools can enrich campus activities and create a harmonious, friendly, and positive school climate, thereby enhancing adolescences’ sense of participation in school activities, their sense of belonging to the school, and their identification with the school’s values [77].

### 4.4. Limitations and Future Directions

First, this study used a convenience sample, which may have made the sample less representative and affected the generalizability of the results. Because son preference is a phenomenon that is especially severe in specific geographical locations, such as rural areas and West and Central China, results from the use of a sample city only in western China cannot be generalized to a broader geographical and cultural background. Therefore, future research is encouraged to expand the selection of participants. Second, this study used a cross-sectional approach to explore the relationship between variables which could not reflect the causal relationship between variables well. Therefore, a longitudinal study should be used in the future to explore the dynamic relationship between maternal son preference, parent–child attachment, school connectedness, and adolescent depressive symptoms. Third, due to the low detection rate of depressive symptoms, further research is needed to determine the applicability of the results to adolescents with significant depression. Fourth, the direct effect in the mediation model is relatively small, which may be significant due to the large sample size, so more studies are needed to verify this model in the future. Finally, this study only focused on the influence mechanism of maternal son preference on adolescent depressive symptoms. Although it is generally believed that maternal son preference reflects a family’s preference for gender, there may also be differences between maternal son preference and paternal son preference. Future research should also focus on paternal son preference and explore the influence of both maternal son preference and paternal son preference on adolescent depressive symptoms. And considering that only adolescent-rated parental son preference may bring skewness to the results, future studies may use a multiple-informant method to combine both parent reports and adolescent reports.

## 5. Conclusions

This study uses a relatively large sample to address the question “how is maternal son preference associated with adolescents’ depressive symptoms?”. This provides new empirical evidence for the causes of adolescent depressive symptoms and provides new insights into improving parenting and family interactions. Specifically, maternal son preference indirectly associates with adolescent girls’ depressive symptoms via impairing mother–daughter attachment and father–daughter attachment. In addition, a low level of school connectedness enhances the risk of girls’ depressive symptoms when they experience maternal son preference, while a high level of school connectedness buffers the risk of girls’ depressive symptoms when they experience maternal son preference. In addition, a high level of school connectedness is also a protective factor for boys’ depressive symptoms when they are preferred by their mother. The results indicate that more attention should be paid to the phenomenon of son preference and gender inequality in family systems to guarantee adolescent girls’ psychological well-being and healthy development.

## Figures and Tables

**Figure 1 behavsci-14-00104-f001:**
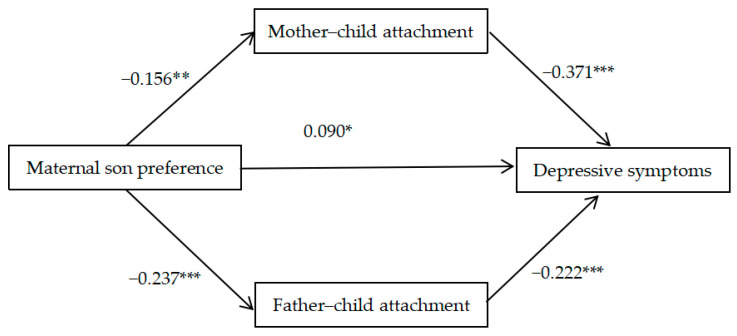
The mediating effect of parent–child attachment in maternal son preference and depressive symptoms among adolescent girls. Note: * *p* < 0.05, ** *p* < 0.01, *** *p* < 0.001.

**Figure 2 behavsci-14-00104-f002:**
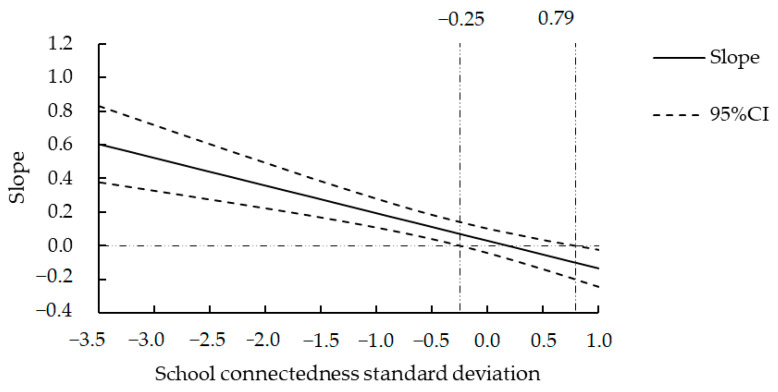
The moderating effect of school connectedness on maternal son preference and depressive symptoms among adolescent girls.

**Figure 3 behavsci-14-00104-f003:**
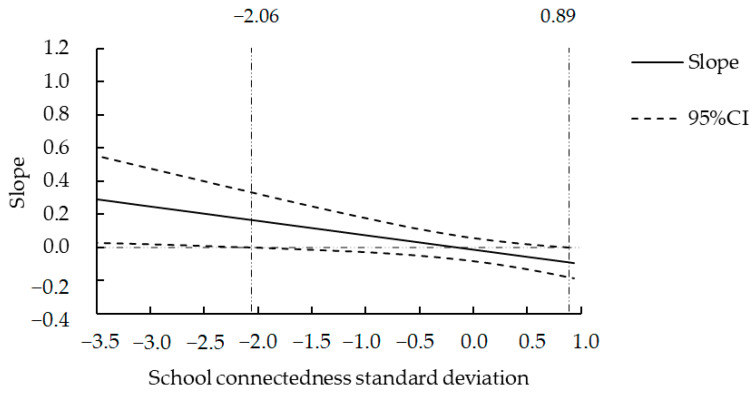
The moderating effect of school connectedness on maternal son preference and depressive symptoms among adolescent boys.

**Table 1 behavsci-14-00104-t001:** Descriptive statistics and correlation analysis among adolescent boys (*n* = 631).

	M ± SD	1	2	3	4	5
1. Maternal son preference	0.86 ± 0.88	1				
2. Mother–child attachment	3.90 ± 0.64	−0.06	1			
3. Father–child attachment	3.84 ± 0.72	−0.02	0.68 ***	1		
4. School connectedness	4.39 ± 0.65	−0.09 *	0.50 ***	0.47 ***	1	
5. Depressive symptoms	0.62 ± 1.06	0.02	−0.45 ***	−0.41 ***	−0.47 ***	1

Note: * *p* < 0.05, *** *p* < 0.001.

**Table 2 behavsci-14-00104-t002:** Descriptive statistics and correlation analysis among adolescent girls (*n* = 462).

	M ± SD	1	2	3	4	5
1. Maternal son preference	0.19 ± 0.44	1				
2. Mother–child attachment	3.90 ± 0.61	−0.16 ***	1			
3. Father–child attachment	3.79 ± 0.68	−0.24 ***	0.71 ***	1		
4. School connectedness	4.43 ± 0.57	−0.13 **	0.47 ***	0.50 ***	1	
5. Depression symptoms	0.61 ± 0.96	0.20 ***	−0.54 ***	−0.51 ***	−0.54 ***	1

Note: ** *p* < 0.01, *** *p* < 0.001.

**Table 3 behavsci-14-00104-t003:** The mediating effect of parent–child attachment.

Effect Type	Effect Size	Boot SE	Bootstrap 95% CI		Relative Effect Ratio (%)
			LLCI	ULCI	
Total effect	0.201	0.046	0.111	0.290	100
Direct effect	0.090	0.039	0.013	0.167	44.8
Indirect effect 1	0.058	0.027	0.013	0.118	28.9
Indirect effect 2	0.053	0.023	0.014	0.103	26.3

Note: Indirect effect 1—maternal son preference → mother–child attachment → depressive symptoms; indirect effect 2—maternal son preference → father–child attachment → depressive symptoms.

**Table 4 behavsci-14-00104-t004:** The moderating effect of school connectedness on maternal son preference and girls’ depressive symptoms.

Dependent Variable	Independent Variable	*β*	*SE*	*t*	*R* ^2^	*F*
MCA	MSP	−0.087	0.043	−2.026	0.233	46.297 ***
	SC	0.454	0.041	10.947		
	MSP × SC	0.036	0.038	0.962		
FCA	MSP	−0.158	0.042	−3.804	0.283	60.377 ***
	SC	0.470	0.040	11.714		
	MSP × SC	0.058	0.036	1.605		
Depressive symptoms	MSP	0.030	0.037	0.805	0.458	54.736 ***
	MCA	−0.303	0.051	−5.897 ***		
	FCA	−0.076	0.053	−1.437		
	SC	−0.303	0.043	−6.972 ***		
	MSP × SC	−0.164	0.034	−4.829 ***		
	MCA × SC	−0.027	0.043	−0.614		
	FCA × SC	0.076	0.046	1.658		

Note: *** *p* < 0.001. MSP = maternal son preference; MCA = mother–child attachment; FCA = father–child attachment; SC = school connectedness.

## Data Availability

The data that support the findings of this study are contained within the article and are available from the corresponding author upon reasonable request.

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
