# Peer review of "Gendered Parenting: Maternal Son Preference and Depressive Symptoms in Chinese Early Adolescents"

_behavsci, 2024, doi:10.3390/bs14020104_

Round 1
Reviewer 1 Report
Comments and Suggestions for Authors
This paper was well-done and presents important results. There are some suggestions for improvement:
1) In the Abstract, the study shows "how" but not "when" about correlates of maternal son preference, as it is not a longitudinal study. The "when" should be removed.
2) The Abstract should explain that the mediation effect was found only for girls.
3) The first paragraph in Section 1.1 belongs in the previous section, as it does not address adolescent depression.
4) The description of family systems theory should have a citation.
5) The claims in the bottom paragraph of page 3 should be supported with citations.
6) In Section 1.3, the "interconnected subsystems" in ecological theory form their own system, termed the "mesosystem." This term should be used.
7) For hypotheses 1 and 2, why would results only be expected for girls and not sons?
8) In Section 2.1, what were the grade level percentages for the students? What were school characteristics, such as size? What was the range of the scale for socioeconomic status?
9) All measures should give example items for each subscale.
10) The beginning of Section 3.1 shows that only around 14% of the sample had any depression. Thus, these results do not apply directly to students experiencing significant depression. This fact should be noted as a limitation.
11) Mediation and moderation analyses should be conducted for sons as well as girls, as just because maternal son preference and depressive symptoms are not correlated, there could be full mediation and/or an interaction term for school connectedness that is significant.
12) The direct effect in the mediation model is .09, which is small in magnitude and likely only significant due to the large sample size. This should be noted.
13) The theories presented in Section 4.1 should initially be presented in the Introduction on maternal son preference and then used to frame results.
14) A potential limitation is that adolescents rated maternal son preference and not the mothers themselves, which could skew the results.
15) The generalizability limitation to other geographic locations and cultures should be mentioned. Using a convenience sample is also a generalizability limitation.
16) At the end of the paper, implications for improved parenting and family interactions would add to the paper.
Comments on the Quality of English LanguageThe term "convenient sample" is used in place of the correct term "convenience sample."
Author Response
Comments 1: In the Abstract, the study shows "how" but not "when" about correlates of maternal son preference, as it is not a longitudinal study. The "when" should be removed.
Response: Thank you for pointing this out. We have made corresponding revisions in the manuscript. Please refer to line 2, page 1.
Comments 2: The Abstract should explain that the mediation effect was found only for girls. |
Response: Thank you for pointing this out. We agree with this comment and have modified the explanation of the mediating effect in the abstract to say that “mother-child attachment and father-child attachment mediated the relationship between maternal son preference and girls’ depressive symptoms”. Please refer to line 9, page 1.
Comments 3: The first paragraph in Section 1.1 belongs in the previous section, as it does not address adolescent depression. Response: Thanks for the comments. We have made some adjustments to the introduction to make it more relevant to the next paragraph. The overall logic is as follows: Mothers have different attitudes and investments to children of different genders, mothers' preference for boys reduces girls' resources (resource dilution theory) and girls' self-worth (social comparison theory), thus affecting girls' mental health. Please refer to line 14-26, page 2.
Comments 4: The description of family systems theory should have a citation. Response: Thanks for the comments. We have added a reference related to family systems theory. Please refer to line 17, page 3. Reference: Cox, M. J., & Paley, B. (1997). Families as systems. Annual Review of Psychology, 48, 243–267. https://doi.org/10.1146/annurev.psych.48.1.243
Comments 5: The claims in the bottom paragraph of page 3 should be supported with citations. Response: Thanks for the comments. We have added some references in this section. Please refer to line 8, page 4. References: Sheehan, G., & Noller, P. (2002). Adolescents' perceptions of differential parenting: Links with attachment style and adolescent adjustment. Personal Relationships, 9(2), 173-190. https://doi.org/10.1111/1475-6811.00011 Liu, Y. W., Su, Y. T., & Yin, Y. R. (2022). Parental preference for boys in childhood and the health of the elderly: Evidence from China. Social Science & Medicine, 302, Article 114986. https://doi.org/10.1016/j.socscimed.2022.114986
Comments 6: In Section 1.3, the "interconnected subsystems" in ecological theory form their own system, termed the "mesosystem." This term should be used. Response: Thanks for the comments. We have made corresponding revisions in the manuscript and added a reference regarding ecological systems theory. Please refer to Section 1.3, lines 15-18, page 4. Reference: Bronfenbrenner, U. (1979). The ecology of human development: Experiments by nature and design.Harvard University Press.
Comments 7: For hypotheses 1 and 2, why would results only be expected for girls and not sons? Response: Thanks for the comments. In the introduction part, we added relevant research on parental preference and adolescent depressive symptoms, and find that previous studies have not reached consistent results. Please refer to Section 1.1, lines 1-5 of paragraph 3, Page 2. Based on this, we do not make specific assumptions about maternal boy preference and depressive symptoms in male adolescents, and we account for this in our assumptions (hypotheses 1). For hypotheses 2, we have modified the hypothesis, expected results in girls and boys are established (hypotheses 2). References: Jeannin, R., & Van Leeuwen, K. (2015). Associations Between Direct and Indirect Perceptions of Parental Differential Treatment and Child Socio-Emotional Adaptation. Journal of Child and Family Studies, 24(6), 1838–1855. https://doi.org/10.1007/s10826-014-9987-3 Wang, M. Z., Wang, H., Guang, S. K., Yan, Y. J., Zhao, Y., & Wang, J. (2019). Perceived parental disfavoritism and life satisfaction in Chinese adolescents: Belief in a just world as the mediator and child gender as the moderator. Children and Youth Services Review, 99, 279–285. https://doi.org/10.1016/j.childyouth.2019.02.017
Comments 8: In Section 2.1, what were the grade level percentages for the students? What were school characteristics, such as size? What was the range of the scale for socioeconomic status? Response: Thanks for the comments. All the students are from grade one in a junior high school. The school is a private school located in Xi’an, a relatively rich city in western China. It covers an area of 500 acres. In this school, it has around 24 grade one classes. In this study, we chose students from 21 classes to voluntarily participate in this study. Please refer to Section 2.1, lines 2-4+11-12, Page 5. Subjective socioeconomic status: This study used the Mac Arthur Scale of subjective socioeconomic status, designed by Adler et al. (2000). During the measurement, participants were presented with a 10-step ladder and asked to assess which rung on the ladder they perceived themselves to belong to. Subjective socioeconomic status ranges from 1 to 10. Reference: Adler, N. E., Epel, E. S., Castellazzo, G., & Ickovics, J. R. (2000). Relationship of subjective and objective social status with psychological and physiological functioning: Preliminary data in healthy white women. Health Psychology, 19(6), 586-592. https://doi.org/10.1037/0278-6133.19.6.586
Comments 9: All measures should give example items for each subscale. Response: Thanks for the comments. We have added examples for each subscale in the manuscript. Please refer to Section 2.2, pages 5-6.
Comments 10: The beginning of Section 3.1 shows that only around 14% of the sample had any depression. Thus, these results do not apply directly to students experiencing significant depression. This fact should be noted as a limitation. Response: Thanks for the comments. We have incorporated this point into the limitations section of the manuscript. Please refer to lines 10-14, Page 10.
Comments 11: Mediation and moderation analyses should be conducted for sons as well as girls, as just because maternal son preference and depressive symptoms are not correlated, there could be full mediation and/or an interaction term for school connectedness that is significant. Response: Thanks for the comments. In the results, we added analyses of mediating and moderating effects between maternal son preference and depressive symptoms in male adolescents. The results showed that the mediating effect of parent-child attachment was not significant, but school connectedness moderated the relationship between maternal son preference and depressive symptoms in male adolescents. Please refer to lines 12-16, Page 10 and lines 1-16, page 11.
Comments 12: The direct effect in the mediation model is .09, which is small in magnitude and likely only significant due to the large sample size. This should be noted. Response: Thanks for the comments. We will address this issue in the limitations of the manuscript. Please refer to lines 12-14, Page 14.
Comments 13: The theories presented in Section 4.1 should initially be presented in the Introduction on maternal son preference and then used to frame results. Response: Thanks for the comments. In the introduction, we have added the relevant content of resource dilution theory and social comparison theory. Please refer to Section 1.1, lines 7-9, page 2 and lines 6-10 of paragraph 2, Page 2. References: Blake, J. (1981). Family size and the quality of children. Demography, 18(4), 421–442. https://doi.org/10.2307/2060941 Buist, K. L., Dekovic, M., & Prinzie, P. (2013). Sibling relationship quality and psycho-pathology of children and adolescents: A meta-analysis. Clinical Psychology Review, 33(1), 97–106. Boyle, M. H., Jenkins, J. M., Georgiades, K., Cairney, J., Duku, E., & Racine, Y. (2004).Differential-maternal parenting behavior: Estimating within- and between-family effects on children. Child Development, 75(5), 1457–1476. Jensen, A. C., Pond, A. M., & Padilla-Walker, L. M. (2015). Why can’t I be more like my brother? The role and correlates of sibling social comparison orientation. Journal of Youth and Adolescence, 44, 2067–2078.
Comments 14: A potential limitation is that adolescents rated maternal son preference and not the mothers themselves, which could skew the results. Response: Thank you for your careful review and insightful feedback. We totally agree with you that the report from mothers themselves is also necessary. Considering that mother report of son preference may be influenced by social desirability and adolescents’ perception of mother’s attitudes may be a more direct factor influencing their development, we finally decided to use adolescent report. We believe that future studies may use multiple informant method to combine both parent report and adolescent report.
Comments 15: The generalizability limitation to other geographic locations and cultures should be mentioned. Using a convenience sample is also a generalizability limitation. Response: Thank you for your thoughtful review and insightful comments. In the revised manuscript, we explicitly addressed these limitations in the limitations section. We will emphasize that the findings should be interpreted within the context of the specific geographic location and cultural context in which the study was conducted. Please refer to lines 2-6, Page 14.
Comments 16: At the end of the paper, implications for improved parenting and family interactions would add to the paper. Response 16: Thanks for the comments, we have revised it in the manuscript. Please refer to Section 5, lines 3-4, Page 14. |

Reviewer 2 Report
Comments and Suggestions for Authors
I would like to thank you for your valuable contribution to the field. Your manuscript adds new empirical evidence for the causes of adolescent depressive symptoms. I recommend paper acceptance, after address some issues below.
Your introduction provides a solid foundation for your study. However, I would like to see an argument to support that the family is the most important microenvironment for adolescents (1.1. at the beginning of the second paragraph). Additionally, please add a reference to enhance the argument about the family system theory (1.2. at the beginning of the second paragraph). I would also suggest a reference to the contrast between the importance of the family as a microsystem (as mentioned in 1.1) and the adolescents’ experience of a gradual detachment from their family environments during the transitional phase of middle school (1.3. at the beginning of the first paragraph).
Please clarify what kind of gift each participant received to participate and explain why this was necessary (2.2).
The information included in the statistical analysis section is very helpful.
The mediation and moderation analyses are the strengths of the paper.
The discussion around the school connectedness as a protective factor is insightful. However, the authors could expand on how schools and teachers can operationalize this knowledge to better support female students (4.3. at the end of the last paragraph).
Author Response
Comments 1: I would like to see an argument to support that the family is the most important microenvironment for adolescents (1.1. at the beginning of the second paragraph). |
Response 1: Thank you for pointing this out. We have included ecological systems theory in this paragraph to support this view. The detailed revision (please see Line 28 to 31 in page 2) is as follows: According to ecological systems theory, family is the most important microenvironment for adolescents, as it is the immediate setting of adolescents’ daily interactions, relationships, and it exerts a direct and profound impact on adolescents’ development (Bronfenbrenner, 1979). Therefore, parents’ beliefs and attitudes toward their children directly affect their growth and development.
|
Comments 2: Please add a reference to enhance the argument about the family system theory (1.2. at the beginning of the second paragraph) |
Response 2: Thank you for pointing this out. We have added a reference related to family systems theory (page 3: line 17). Reference: Cox, M. J., & Paley, B. (1997). Families as systems. Annual Review of Psychology, 48, 243–267. https://doi.org/10.1146/annurev.psych.48.1.243
Comments 3: I would also suggest a reference to the contrast between the importance of the family as a microsystem (as mentioned in 1.1) and the adolescents’ experience of a gradual detachment from their family environments during the transitional phase of middle school (1.3. at the beginning of the first paragraph). Response 3: Thanks for the comments. We have added the references and revised the corresponding parts of the manuscript. The detailed revision (please see line 15 to 17 in page 4) is as follows: During the transitional phase of middle school, adolescents experience a gradual detachment from their family environments and spend less time interacting with their families and more time within the school environment. Reference: Lin, C. D.(2018). Developmental psychology. Peking University Press.
Comments 4: Please clarify what kind of gift each participant received to participate and explain why this was necessary (2.2) Response 4: Thanks for the careful comments. We checked the investigation note and found that we made a mistake to say “each participant received a small gift”. We usually give gifts to participants as a reward for their kind participation in our relatively long investigation. But during this investigation, we did not give gifts, instead, we gave feedback to the school and teachers in responsible about the overall situation of students’ family, school environment and mental health. Therefore, we deleted this sentence in our manuscript. Thanks again for your careful work.
Comments 5: The discussion around the school connectedness as a protective factor is insightful. However, the authors could expand on how schools and teachers can operationalize this knowledge to better support female students (4.3. at the end of the last paragraph). Response 5: Thanks for the comments. In the part of data analysis, this study increased the moderating role of school connectedness between maternal son preference and depressive symptoms in male adolescents (according to reviewer 1’s comments), and found that school connectedness also played a protective role in male adolescents’ mental health, as higher level of school connectedness can buffer the effect of maternal son preference on boys’ depressive symptoms. Therefore, when proposing interventions, there is no distinction between adolescents of specific gender. We have added to the manuscript three ways that may improve school connectedness. First, in enhancing peer support, teachers and schools can organize peer support groups to promote interaction and cooperation among students, so that adolescents can seek support from their classmates when they encounter difficulties (Eccles & Roeser, 2011). Second, in enhancing teacher support, teachers should adopt supportive communication methods and pay timely attention to students' individual needs, so as to establish a positive teacher-student relationship and enhance students' trust in teachers (Roorda et al., 2017). Third, schools can enrich campus activities and create a harmonious, friendly, and positive school climate, thereby enhancing adolescences’ sense of participation in school activities, their sense of belonging to the school, and their identification with the school's values (Wang & Degol, 2016). Please refer to Section 4.3, paragraph 4, lines 4-13. References: Eccles, J. S., & Roeser, R. W. (2011). Schools as Developmental Contexts During Adolescence. Journal of Research on Adolescence, 21(1), 225–241. https://doi.org/10.1111/j.1532-7795.2010.00725.x Roorda, D. L., Jak, S., Zee, M., Oort, F. J., & Koomen, H. M. Y. (2017). Affective Teacher-Student Relationships and Students' Engagement and Achievement: A Meta-Analytic Update and Test of the Mediating Role of Engagement. School Psychology Review, 46(3), 239–261. https://doi.org/10.17105/spr-2017-0035.V46-3 Wang, M. T., & Degol, J. L. (2016). School Climate: a Review of the Construct, Measurement, and Impact on Student Outcomes. Educational Psychology Review, 28(2), 315–352. https://doi.org/10.1007/s10648-015-9319-1
|
